# Near-term forecasting of companion animal tick paralysis incidence: An iterative ensemble model

Nicholas J. Clark[1]*, Tatiana Proboste[1], Guyan Weerasinghe[2], Ricardo J. Soares Magalhães[1,3]

1 UQ Spatial Epidemiology Laboratory, School of Veterinary Science, the University of Queensland, Gatton, Queensland, Australia, 2 Department of Agriculture, Water and the Environment, Canberra, Australia, 3 Children's Health and Environment Program, UQ Child Health Research Centre, the University of Queensland, Gatton, Australia

* n.clark@uq.edu.au

**Data Availability Statement:** All clinical and environmental time series data, along with R code to reproduce ensemble modelling and particle filters, is provided in S1 Data and is permanently

## Abstract

Tick paralysis resulting from bites from *Ixodes holocyclus* and *I. cornuatus* is one of the leading causes of emergency veterinary admissions for companion animals in Australia, often resulting in death if left untreated. Availability of timely information on periods of increased risk can help modulate behaviors that reduce exposures to ticks and improve awareness of owners for the need of lifesaving preventative ectoparasite treatment. Improved awareness of clinicians and pet owners about temporal changes in tick paralysis risk can be assisted by ecological forecasting frameworks that integrate environmental information into statistical time series models. Using an 11-year time series of tick paralysis cases from veterinary clinics in one of Australia's hotspots for the paralysis tick *Ixodes holocyclus*, we asked whether an ensemble model could accurately forecast clinical caseloads over near-term horizons. We fit a series of statistical time series (ARIMA, GARCH) and generative models (Prophet, Generalised Additive Model) using environmental variables as predictors, and then combined forecasts into a weighted ensemble to minimise prediction interval error. Our results indicate that variables related to temperature anomalies, levels of vegetation moisture and the Southern Oscillation Index can be useful for predicting tick paralysis admissions. Our model forecasted tick paralysis cases with exceptional accuracy while preserving epidemiological interpretability, outperforming a field-leading benchmark Exponential Smoothing model by reducing both point and prediction interval errors. Using online particle filtering to assimilate new observations and adjust forecast distributions when new data became available, our model adapted to changing temporal conditions and provided further reduced forecast errors. We expect our model pipeline to act as a platform for developing early warning systems that can notify clinicians and pet owners about heightened risks of environmentally driven veterinary conditions.

archived in Figshare (https://doi.org/10.6084/m9.figshare.16920838.v1). Raw clinic-level data could potentially identify or reveal sensitive patient information. Public deposition would breach compliance with our approved ethics protocol These data can only be made available upon formal request from the Chief Veterinary Officer of Greencross Australia (magdoline.awad@gxltd.com.au, https://www.greencrossvets.com.au/).

**Funding:** This study was funded by an ARC DECRA Fellowship to NJC (DE210101439). The funders had no role in study design, data collection and analysis, decision to publish, or preparation of the manuscript.

**Competing interests:** The authors have declared that no competing interests exist.

## Author summary

Tick-borne illnesses constitute a diverse group of debilitating conditions for pet dogs and cats around the world. In Australia, thousands of domestic dogs are admitted to emergency veterinary clinics due to tick paralysis each year. These admissions are highly seasonal and may be associated with changing environmental conditions, suggesting models that learn from environmental patterns to forecast the oncoming tick season could inform pet owners and clinicians about changing risks. In this paper we use a series of statistical forecasting models to analyse and predict tick paralysis admissions to veterinary clinics in a tick paralysis hotspot in Queensland, Australia. Our approach is novel in that we combine individual models into a superior ensemble that is trained to reduce forecast uncertainty, giving more accurate estimates of what the coming tick season will look like. Our model consistently outperforms a field-leading benchmark while uncovering important patterns about environmental drivers of paralysis tick exposure, including changes to levels of moist vegetation and maximum temperature. We also demonstrate how our model can be used to automatically produce forecasts of tick paralysis admissions as new data become available. This can have important implications for designing improved early warning systems for tick-borne illness.

## Introduction

Tick paralysis, caused by neurotoxins in the saliva of ixodid hard ticks (Family Ixodidae; Genus *Ixodes*), is a life-threatening vector-borne disease affecting a range of vertebrates including humans [1,2]. Dogs and cats are the most vulnerable and frequently parasitized domestic host species, with paralysis tick envenomation in these animals requiring immediate veterinary medical attention [3,4]. Symptoms of tick paralysis include loss of appetite, poor mobility, lack of coordination and severe respiratory problems [5]. The condition is often fatal if untreated. Prevention of paralysis tick bites relies on prophylactic acaricide drugs, equating to millions of dollars spent by pet owners each year [6]. Reducing exposures to ticks is crucial to limit envenomation risk. This need is heightened by evidence that paralysis ticks can cause severe allergic conditions in domestic animals and humans and can act as vectors for a suite of zoonotic pathogens (including Rickettsia, Orienta, Bartonella and Babesia species; [7,8]).

In Australia, paralysis caused by bites from *Ixodes holocyclus* and *I. cornuatus* is a leading cause of emergency admissions for dogs, with tens of thousands of cases admitted each year [9,10]. Both tick species are distributed along Australia's east coast and are associated with wet and humid habitats [11], though *I. holocyclus* is more widespread and more commonly implicated in tick paralysis cases [8]. Paralysis tick distributions coincide with some of Australia's most populous urban centers, placing millions of pets at risk of tick paralysis each year. Despite recent advances in veterinary emergency care and widespread use of preventative treatments [10], avoidance of tick-prone areas during periods of heightened risk remains a key control measure. Ecological studies have provided vital information on the life histories, distributions and habitat preferences of *Ixodes* ticks in Australia [12–14]. Environmental measurements such as temperature, vegetation structure or relative moisture can lead to better predictive capacity and help uncover new insights into the epidemiology of vector-borne illnesses [15]. Modelling frameworks that combine this information with clinical data to forecast tick paralysis burdens are now needed to improve awareness about changes in risk.

Identifying ways to improve forecasts of vector-borne conditions is necessary to bolster awareness, manage resources and understand risk [16–18]. Because disentangling the

processes that drives variation in a time series is often challenging, combining multiple forecasts can "hedge against the risk of selecting a mis-specified model" [19]. Studies from diverse fields have shown that combining forecasts into weighted ensembles improves prediction [20–25]. It is reasonable to assume that tick paralysis admissions are amenable to ensembles as some aspects of clinical admissions should be highly predictable by simple forecast models (e.g. strong seasonality in tick feeding patterns) while others call for models that learn from key environmental covariates (e.g. associations with climate and vegetation conditions, dependence on host population fluctuations). But selecting weights for combining forecasts into ensembles is difficult. Machine learning approaches that treat the combination as a minimization problem can overcome this hurdle, albeit at the potential cost of reduced interpretability [19]. Bridging the gap between near-term prediction and paralysis tick prevention requires studies that build ecologically relevant models and explore how their uncertainties can be reduced to make forecasts more useful and interpretable.

This study aimed to develop and test a tick paralysis ensemble forecasting model which integrates information from high-resolution environmental measurements to address the need for improved prediction of paralysis tick burdens in domestic dogs. Our methodology provides a framework for automated building and updating of forecast distributions that can be adapted to a range of important veterinary vector-borne diseases of public health importance.

## Materials and methods

### Ethical statement

Ethical clearance for use of these data was provided by the University of Queensland Human Ethics Research Office (approval 2018000514).

### Data on canine tick paralysis cases

Clinical records of paralysis tick cases in domestic pets were sourced from participating Greencross Veterinary clinics in the Sunshine Coast Local Government Area, Queensland, Australia (Fig 1). The Sunshine Coast has one of Australia's highest rates of domestic animal tick paralysis, primarily due to its warm and wet subtropical climate and its high cover of moist shrubland vegetation [14]. We abstracted domestic animal tick paralysis cases from all five clinics for the period 2007–2017, inclusive. Abstraction was performed by prioritizing cases that were either confirmed or probable tick paralysis using a search strategy that examined patient histories, diagnoses and treatments for common variations of the following terms: "paralysis", "ataxia", "paresis", "uncoordination", "incoordination", "tick", "*Ixodes*", "anti serum", "antiserum", "anti-serum", "TAS". This strategy returned a total of 766 records of domestic animal tick paralysis cases (77% were dogs). Some records were duplicates, which included animals admitted multiple times within a short period. After removing duplicates, we retained 520 cases. Inspection of 100 randomly chosen records identified 80 confirmed (tick and/or tick crater found, treated with tick antiserum), 15 probable (suspected paralysis but no tick or crater found, treated with tick antiserum) and five improbable records (e.g. vaccination, discussed preventative treatment for tick paralysis). To allow for automated updating of record retrieval for future forecasts, we retained all records and assumed our identification procedure had ~95% specificity.

Paralysis tick cases were binned into temporal periods to provide adequate sample sizes for model cross-validation while still capturing the strong seasonality in admissions. To ensure temporal units could be directly compared across years, we binned the data into half-months (referred to as 'fortnights' herein) by splitting each calendar month into two evenly sized

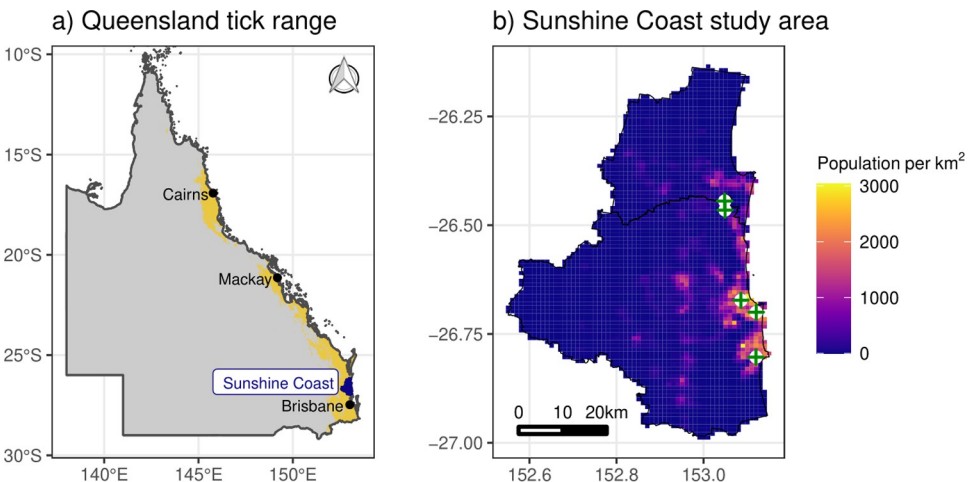

**Fig 1. Location of the study area.** a) Location of the Sunshine Coast study area (blue coloured polygon) within the approximate distribution of the paralysis tick *Ixodes holocyclus* in Queensland, Australia (yellow coloured region). The approximate distribution is the output of a species distribution model trained on open access occurrence records (Atlas of Living Australia, Global Biodiversity Information Facility) and is used here for illustrative purposes only. b) Locations of participating veterinary clinics (cross symbols) within the Sunshine Coast. Colours represent human population density as of 2018. The Queensland state polygon was accessed from the Australian Bureau of Statistics at https://www.abs.gov.au/statistics/standards/australian-statistical-geography-standard-asgs-edition-3/jul2021-jun2026/access-and-downloads/digital-boundary-files/STE_2021_AUST_SHP_GDA2020.zip.

temporal periods (periodicity = 24). We used seasonal-trend decomposition based on Loess smoothing (STL; [26]) to decompose the series into seasonal, trend and remainder components (Fig 2). In an STL decomposition, the seasonal component is calculated by applying

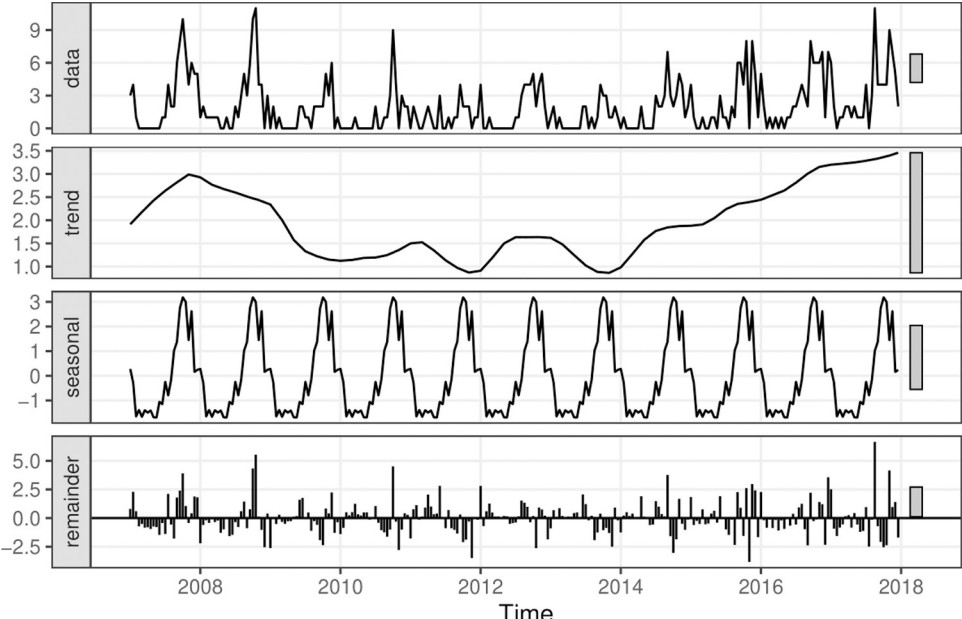

**Fig 2. Seasonal-trend decomposition based on Loess smoothing of paralysis tick admissions.** The raw time series (shown in the data panel at the top) was decomposed into trend, seasonality and remainder components. Our unit of analysis for two prediction models (ARIMA$_{seasadj}$ and GARCH$_{seasadj}$) was the seasonally adjusted series, taken as the sum of the trend and remainder. Remaining models used either the raw count data (GAM$_{raw}$) or a *log(x+1)* transformation of the raw data as the outcome (Prophet$_{raw}$, and ETS$_{raw}$). Bars in panels represent relative differences in y-axis scales.

Loess smoothing to the seasonal sub-series, which in our case meant smoothing over the 24 fortnights. After seasonality was removed, the non-seasonal component was Loess smoothed to find the trend, which represents the long-term progression of the series. The remainder component is the residuals from the seasonal plus trend fit. Because our data demonstrated consistent seasonality (Fig 2) that violates stationarity assumptions required by some time series models, we fit some of our prediction models to the seasonally adjusted series (the sum of the trend and the remainder components in Fig 2). A Dickey-Fuller test for whether the seasonally adjusted series had a unit root (i.e. testing against the hypothesis that a series is not stationary) confirmed stationarity ($t = -3.60$, $p = 0.04$). The seasonally adjusted series was therefore deemed appropriate for models such as the ARIMA and GARCH when attempting to identify predictors of tick cases that were above or below the seasonal average.

## Predictor variables

The distributions and activity levels of paralysis ticks are known to respond to local climate and environmental conditions [11,27,28]. We extracted remote-sensed measurements for nine variables that reflect variation in climate, vegetation, moisture and landcover, all of which can impact paralysis tick ecology. Included variables were maximum temperature, minimum temperature, evapotranspiration, total rainfall, the Normalized Difference Vegetation Index (NDVI) and the proportions of the study area classified as shrubland or forest (sources, spatial and temporal resolutions of predictors shown in Table 1). As paralysis ticks are hypothesized to achieve high densities and increased activity in moist shrub habitats [11,29], we calculated a 'moist vegetation' variable to reflect increasing levels of moisture (more rainfall, less evapotranspiration) and the relative covers of shrub and forest:

$$moist\ vegetation = rainfall * (-1 * evapotranspiration) * (shrub\ cover + forest\ cover)$$

**Table 1. Sources, spatial and temporal resolutions of covariates used to calculate predictors.**

| Variable (unit) | Source | Spatial resolution | Temporal resolution |
|---|---|---|---|
| Maximum temperature (˚C) | Queensland Long Paddock SILO database of Australian climate data[1] | 5km | Daily |
| Minimum temperature (˚C) | Queensland Long Paddock SILO database of Australian climate data[1] | 5km | Daily |
| Total rainfall (˚C) | Queensland Long Paddock SILO database of Australian climate data[1] | 5km | Daily |
| Evapotranspiration (ET$_o$) | Queensland Long Paddock SILO database of Australian climate data[1] | 5km | Daily |
| Southern Oscillation Index (SOI; unitless) | Queensland Long Paddock database of Southern Oscillation Index data[2] | Global | Daily |
| Normalized Difference Vegetation Index (NDVI; unitless) | Australian Bureau of Meteorology[3] | 27km | Monthly |
| Proportional cover of shrubland (landcover categories 120, 121, 122; %) | The European Union's Earth Observation Programme[4] | 300m | Yearly |
| Proportional cover of open or dense forest (landcover categories 40, 50, 60, 61, 62, 71, 72, 80, 81, 82, 90; %) | The European Union's Earth Observation Programme[4] | 300m | Yearly |

[1] Accessed at: https://www.longpaddock.qld.gov.au/silo/ [31,32]

[2] Accessed at: https://www.longpaddock.qld.gov.au/soi/soi-data-files/ [32]

[3] Accessed at: http://www.bom.gov.au/climate/austmaps/about-ndvi-maps.shtml

[4] Derived from global landcover maps: https://cds.climate.copernicus.eu/cdsapp#!/dataset/satellite-land-cover?tab=overview

We used an Australian Statistical Area Level 2 shapefile obtained from the Australian Bureau of Statistics (https://www.abs.gov.au/AUSSTATS/abs@.nsf/DetailsPage/1270.0.55.001July%202016?OpenDocument) to extract raster variables for the Sunshine Coast. For all environmental rasters, we first calculated average values for the entire Sunshine Coast study area. We then calculated quarterly (3-month periods: Jan–March, April–June, July–Sep and Oct–Dec) anomalies (referred to herein as *variable*$_{Q\text{-anomaly}}$) as additional predictors to describe how conditions compared to long-term averages for that same period. To capture remaining variability in rainfall, we included fortnightly averages of the Southern Oscillation Index (SOI), an index representing differences in barometric pressures over Tahiti and Darwin, Australia [30]. The SOI is commonly used to assess the strength of the El Niño Southern Oscillation, a major climate force that accounts for nearly 25% of Queensland's annual rainfall variability and is suggested to be a useful indicator of vector-borne disease infection [17].

Our full set of predictor variables was: moist vegetation, moist vegetation$_{Q\text{-anomaly}}$, maximum temperature, maximum temperature$_{Q\text{-anomaly}}$, minimum temperature, minimum temperature$_{Q\text{-anomaly}}$, NDVI, NDVI$_{Q\text{-anomaly}}$ and SOI. All predictors were aggregated as fortnightly means across the Sunshine Coast study area. Pairwise collinearities revealed strong positive correlations (Pearson correlation > 0.70) between minimum temperature and maximum temperature, and between NDVI and NDVI$_{Q\text{-anomaly}}$. We therefore removed minimum temperature and NDVI from the candidate set. Lags of remaining predictors were generated to span between 1–6 fortnights (2–12 weeks) to prioritize forecasts at horizons that would allow practitioners time to make informed decisions. For each predictor, we selected the lag that showed the strongest cross-correlation with the seasonally adjusted series for inclusion in model testing. Predictors were scaled to unit variance prior to modelling.

## Modelling framework

We followed a sequential process to train an ensemble model to forecast tick paralysis admissions (Fig 3). The first step involved training individual predictor models on either the seasonally adjusted or raw outcome series. The models we implement were chosen because they make different assumptions about time series evolution, can incorporate both seasonality and effects of predictors and because quantification of forecast uncertainty is straightforward. However, our choice of candidate models does not by any means represent an exhaustive set, and we acknowledge that there are many other potentially useful models for analysing timeseries and generating forecast distributions. All models were trained on 85% of the data (225 fortnight observations representing the training set and covering the 2007–2015 tick seasons) and their forecasts evaluated on the remaining 15% (39 observations, including the final two tick seasons). Next, we combined forecasts from these models into a weighted ensemble to minimize out-of-sample prediction interval error for the 2016 period. We then used this weighted ensemble to forecast the 2017 tick season and compared it to forecasts from a field-leading benchmark model. Finally, we simulated a real-world scenario in which ensemble and benchmark models were re-calibrated to incoming data to adjust forecasts and used a rolling window to compare prediction errors (Fig 3). Each step is explained in detail in the following sections. Interested readers can refer to S1 Table in the Supporting Information for details about predictor model assumptions.

**Modelling seasonally adjusted tick paralysis admissions.** Our first prediction task was to model the seasonally adjusted series. We trialed two models that make different assumptions about relationships between predictors and the outcome. Gaussian error distributions were assumed for both models.

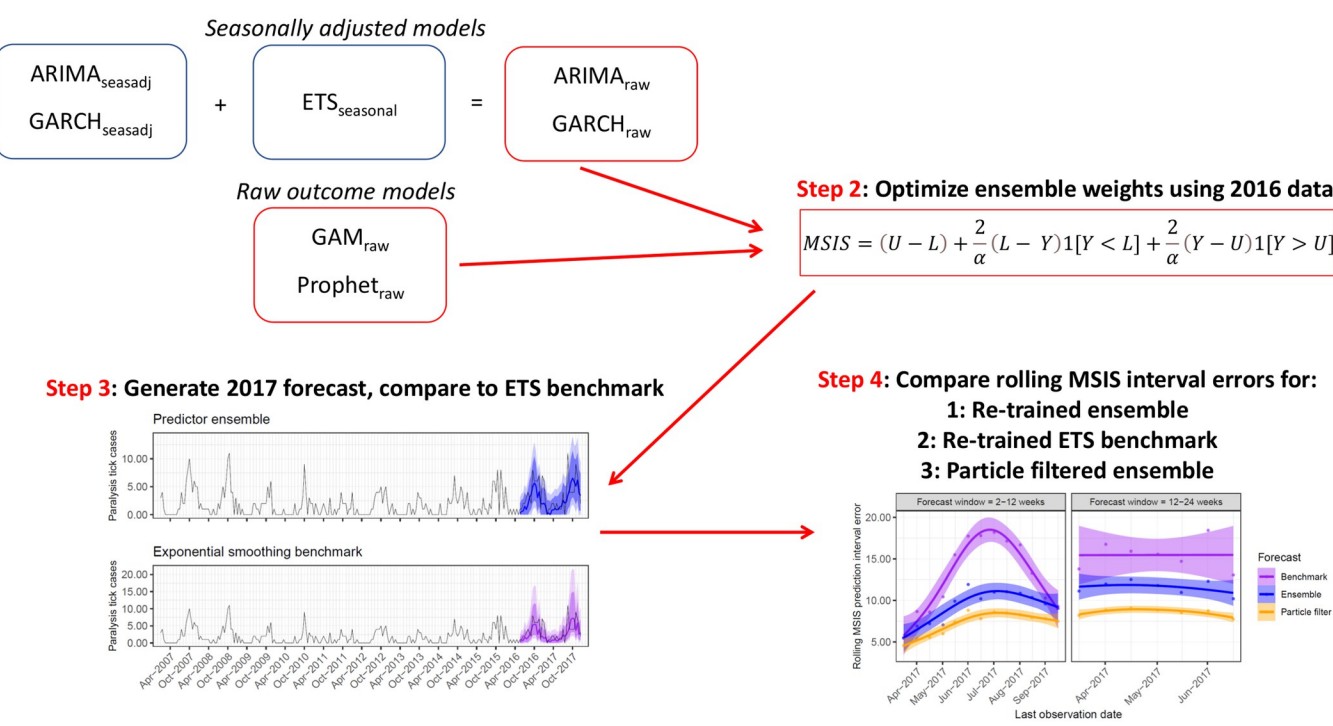

**Fig 3. Schematic representation of the sequential process used to train and interrogate the ensemble forecast for tick paralysis admissions on the Sunshine Coast.** ARIMA, Autoregressive Integrated Moving Average; GARCH; Generalized Autoregressive Conditional Heteroskedasticity; GAM, Generalized Additive Model; ETS, exponential smoothing; MSIS, Mean Scaled Interval Score.

**ARIMA$_{seasadj}$ model.** The first predictor model was an Autoregressive Integrated Moving Average (ARIMA) model. ARIMA models assume the outcome series is stationary so that the autocorrelation coefficients that control how past values predict future values can be applied uniformly throughout the series. We identified environmental predictors and ARIMA parameters (order and moving average coefficients) using a grid search of all candidate models including up to five covariates (we limited the number of predictors to five so that the total number of parameters was not so large as to risk overfitting or overparameterization). For each combination of predictors, we identified the best ARIMA parameters using the "auto. arima" function in the *forecast* R package [33], which optimizes fit by heuristically searching over possible differencing, order and moving average parameters. The final ARIMA$_{seasadj}$ model was selected by finding the ARIMA parameters and predictor set that minimized in-sample Bayesian Information Criterion.

**GARCH$_{seasadj}$.** Our second model was a Generalized Autoregressive Conditional Heteroskedasticity (GARCH) model. A GARCH is appropriate when volatility (sometimes referred to as 'jumpiness') demonstrates an irregular pattern rather than being homogenous over time [34]. When heteroskedasticity occurs, estimates of standard errors and prediction intervals can be biased, impacting performance and inference. GARCH models consider volatility at a given time to be conditional on the volatility at past time points, with changes in variance persisting according to an autoregressive residual moving average structure. To detect whether there was autocorrelation of volatility in the seasonally adjusted series, we fit an auto.arima() model and plotted the autocorrelation function of the absolute residuals. This plot showed significant autocorrelation of volatility up to a lag of two fortnights, justifying the use of the

GARCH model as an appropriate alternative to the ARIMA$_{seasadj.}$ Our chosen model included an ARIMA(0,2) process for the mean, a GARCH(0,2) process for the conditional variance (i.e. both the mean and variance were influenced by two moving average lags) and terms capturing linear additive effects of predictors. Rather than selecting among combinations of predictors as above, we included all predictors simultaneously and placed regularized double exponential priors (mean = 0, sd = 0.5) on their coefficients to capture our prior expectation that most effects of regressors on the seasonally-adjusted series will be small. GARCH parameters were estimated in a Bayesian framework using the No U-Turn Sampler algorithm in *Stan* [35] through the R interface *varstan* [36]. Apart from the regression coefficient priors, all other priors were provided with default distributions. We sampled four chains for 1,000 samples each and gathered 1,000 total samples from the posterior distribution.

**Modelling raw tick paralysis admissions.** We fit two predictor models using the raw tick paralysis series as the outcome variable. This allowed us to determine whether models that can simultaneously capture seasonal, trend and predictor effects performed better than models trained on seasonally adjusted values.

**GAM$_{raw}$.** We fit a Generalized Additive Model (GAM) to the admission count data (GAM-$_{raw}$) assuming admissions were random draws from an unknown Negative Binomial distribution. A Negative Binomial was chosen instead of a Poisson as the Negative Binomial is appropriate for discrete outcomes when excess zeros are present and because our tick admission data displayed discrepancies between the variance and the mean. The Negative Binomial distribution is parameterized by an unknown rate $\lambda$ and overdispersion parameter $\varphi$, where rate at time t is $\lambda_t = \varphi / (\varphi + \mu_t)$. Using a log link, we modelled $\mu$ as an additive combination of smooth functions while capitalizing on automatic variable selection using penalized smooths available in the *mgcv* R package [37]. We included the best lags for each of our covariates as cubic spline smooths. Each smooth incorporated a Bayesian 'shrinkage' prior (bs = 'cs' in *mgcv* notation) that forced coefficients toward zero unless there was likelihood support for their inclusion. According to Marra & Wood [38], this automatic variable selection "can both enhance model interpretability and improve prediction accuracy". We captured the nonlinear trend by including year as an unpenalized cubic spline smooth term, and we included a cyclic cubic regression smooth (bs = 'cc' in *mgcv* notation) for the effect of fortnight to capture seasonality. A moving average autocorrelation structure was used for the residuals. The number and placement of knots for all smooths were automatically chosen using Generalized Cross-validation [39].

**Prophet$_{raw}$.** Our final predictor model was the recently described Prophet model, a modular regression developed by Facebook analysts [40]. Prophet uses a decomposable regression with three primary model components: trend, seasonality and additive linear terms capturing effects of regressors. Gaussian error is used for the residual distribution. The trend is composed of a piecewise linear growth rate that is regularised by a sparse Laplace prior controlling the selection and 'wiggliness' of changepoints. Seasonality is represented using Fourier series to provide a periodic model that can also be regularised by a Gaussian smoothing prior [40]. The nonlinear trend and seasonality components are therefore similar to smooth functions used in GAMs. Prophet has been embraced by the forecasting community, with successful applications in a range of disciplines including predictions of groundwater level [41], electrical loads [42] and business cash flows [43]. Due to Prophet's assumption of Gaussian error, we fit the model to *log(x + 1)* transformed admissions and back-transformed forecasts to the discrete scale. Given the large number of parameters to be estimated and the lack of options for regression coefficient regularisation, we only included five predictors in the Prophet model. The best three predictors were chosen by testing each predictor on its own and finding the three that minimized model deviance. We fit the model using the prophet R package (https://facebook. github.io/prophet/), which uses *Stan* for Bayesian estimation. We used default values for all

parameters apart from the trend changepoint shape prior, which we increased from the default 0.05 to 0.50 to allow greater flexibility in estimating nonlinear trends. As for the GARCH$_{seasadj}$ model above, four chains of 1,000 samples each were run to estimate parameters and 1,000 samples were gathered from the posterior distribution.

**Building a weighted predictor ensemble forecast.** To generate forecasts on the outcome scale for the ARIMA$_{seasadj}$ and GARCH$_{seasadj}$, forecasts of seasonally adjusted counts were added to seasonal forecasts from an exponential smoothing (ETS) model in which trend, seasonality and error components were chosen automatically using the 'ets' function in the *forecast* package (Fig 3; S1 Table). Forecasts from the GAM$_{raw}$ and Prophet$_{raw}$ were produced on the raw scale. Our goal was to find a weighted ensemble of predictor forecasts that could minimize uncertainty without sacrificing accuracy. We calculated weights that minimized the prediction interval using the Mean Scaled Interval Score (MSIS) metric proposed by Gneiting and Raftery [44]. The MSIS metric penalizes intervals that are excessively large and / or that do not retain the true (observed) value:

$$MSIS = (U - L) + \frac{2}{\alpha}(L - Y)\mathbf{1}[Y < L] + \frac{2}{\alpha}(Y - U)\mathbf{1}[Y > U]$$

Here, *U* and *L* represent a weighted average of upper and lower 95% intervals at a particular horizon, *Y* represents the observation at that horizon, the significance level is $\alpha$ (set to 0.05) and **1** is an indicator function (returning a 1 if the evaluated expression is true and 0 otherwise). The value of the equation increases when *Y* is outside the weighted interval due to the penalty terms (i.e. one of the indicator functions will return a 1 when *Y* is not in the interval). We used nonlinear gradient optimization [45] to minimize the MSIS function across the 2016 validation period.

**Comparing to an Exponential Smoothing benchmark forecast.** We compared forecasts for the 2017 tick season from our ensemble model to those from a naïve baseline ETS model that was trained to forecast *log(x + 1)* transformed paralysis tick admission counts. Exponential smoothing is one of the most effective forecast algorithms across a range of applications as it captures error, seasonality and trend components in a state-space framework [46,47]. The ETS was considered a competitive benchmark as such models that only extract information on trends, seasonality and other components from historical values often do incredibly well in forecast competitions, including those for vector-borne disease [18,48]. Comparisons between our ensemble and the ETS benchmark allowed us to increase transparency regarding the performance of our ensemble while providing valuable insights into the temporal periods and conditions in which external predictors can inform better forecasts. Two approaches were used to compare models. First, we calculated point prediction errors for the 2017 tick season using forecasts from models that were trained on all data up to March 2017 (immediately following the end of the 2016 tick season). We then iteratively updated forecasts using a rolling window to continually estimate forecast performances over near-term (2–12 weeks) and medium-term (12–24 weeks) horizons. This involved re-training ensemble members on data up to mid-March 2017, producing weighted forecasts for both horizons, generating extended forecasts from the re-trained benchmark and calculating prediction errors. The process was repeated by extending the window forward in fortnightly intervals to scrutinize how each model would have performed had it been re-trained on the most recent data available leading into the tick season.

## Data assimilation using Sequential Monte Carlo particle filtering

Assimilation of new observations to adapt forecasts is an important step in automated forecasting [49,50]. Particle filters are some of the most popular assimilation algorithms as they are likelihood-based and can efficiently re-ground models to empirical data while exploring

complex non-Gaussian distributions [51]. We used particle filtering for the two models that were fitted to the seasonally adjusted series (ARIMA$_{seasadj}$ and GARCH$_{seasadj}$) by iteratively assimilating observations from the validation set, rather than recalibrating the models using all data up to that point. This involved simulating a set of particles for each model, with each particle representing one possible forecast trajectory. Particles were propagated forward according to estimated model equations, with the paths of all particles collectively formulating the prior forecast distribution. When the next observation (the next fortnight of cases) was available, a Monte Carlo step was run in which the log likelihood of that observation given each particle's unique proposal was calculated and used to weight the particles, where higher likelihoods lead to higher weights. The prior forecast was then updated into a weighted posterior forecast. Particles more representative of the data generating process were therefore given more importance when adapting the forecast [52]. Following a Sequential Monte Carlo algorithm, the assimilation process was repeated in the next timestep to update weights and adjust the forecast. Particle weights were sequentially updated using a condensation algorithm that iteratively multiplied a particle's current likelihood by its previous likelihood [53]. To prevent particle collapse, where a small number of particles makes up most of the weights, we used importance sampling (resampling with replacement in proportion to weights) when effective sample size fell below a threshold:

$$If \left( \frac{1}{sum(weights^2)} \right) < \frac{N}{2}, resample$$

During resampling, particles with high weights were more likely to be oversampled, while those with low weights were likely to be removed. After resampling, particles were 'mutated' by allowing their trajectories to drift according to random walks with small Gaussian noise, providing the models with flexibility to explore new parameter spaces. Weights were then reset to 1 prior to the next iteration. We simulated 5,000 particles for each model and assimilated observations using the same rolling window approach as above. The remaining models (i.e. GAM$_{raw}$ and Prophet$_{raw}$) were updated as above by retraining on the growing dataset to form the particle filtered ensemble, which we compared to the re-trained ensemble and benchmark models described above.

To further explore our ensemble's capability, we assessed how the particle filter could accommodate spatial variation in time series dynamics. We grouped veterinary clinics by geographical location (one group containing the two northernmost clinics, one group for the two mid-latitude clinics and a third group for the single southernmost clinic; Fig 1). The three datasets showed similar seasonalities but different trends, offering an opportunity to test the particle filter's adaptive capability without the need to parameterize a complex spatio-temporal forecast model. The particle filter was adapted to each clinic-level series from the beginning of 2007. All parameter distributions from the ensemble model were retained for the clinic-level particle filters with the exception of Gaussian error parameters, which were initialized separately for each clinic-level series using estimates from ARIMA(1,0,1) models. Forecasts were then compared to clinic-level ETS benchmarks as above. Data and R code to replicate ensemble modelling and particle filtering steps for the aggregated timeseries is provided in Data S1.

## Results

### Temporal variation in tick paralysis admissions

Paralysis tick admissions tended to peak between September and November, though the range of dates during which admissions were recorded varied across time with no apparent shift in seasonality (Fig 2; S1 Fig). August showed the largest range of admissions, with a minimum of

**Table 2. Summary statistics for predictor coefficients.** All predictors were scaled to unit variance prior to modelling. Significant effects are highlighted in bold. Note that GAM F statistics only indicate the nonlinearity of the estimated smooth function, not the directionality of the effect. ARIMA, Autoregressive Integrated Moving Average; GARCH; Generalized Autoregressive Conditional Heteroskedasticity; GAM, Generalized Additive Model. SE, standard error; CI, Bayesian credible interval.

| Predictor variable | Lag time (fortnights; weeks) | ARIMA$_{seasadj}$ coefficient (SE) | GARCH$_{seasadj}$ coefficient (CI) | GAM$_{raw}$ F statistic ($p$-value) | Prophet coefficient (CI) |
|---|---|---|---|---|---|
| Moist vegetation | 6; 12 | - | 0.08 (-0.15, 0.38) | 0.00 (1.00) | **0.05 (0.01, 0.08)** |
| Moist vegetation$_{Q\text{-anomaly}}$ | 2; 4 | 0.02 (0.08) | -0.12 (-0.33, 0.05) | 0.00 (1.00) | 0.01 (-0.03, 0.05) |
| Maximum temperature | 5; 10 | - | -0.22 (-0.05, 0.36) | 0.22 (0.25) | -0.01 (-0.07, 0.06) |
| Maximum temperature$_{Q\text{-anomaly}}$ | 2; 4 | - | **-0.37 (-0.69, -0.01)** | **3.59 (0.03)** | **-0.04 (-0.07, -0.02)** |
| NDVI$_{Q\text{-anomaly}}$ | 4; 8 | - | -0.11 (-0.54, 0.20) | 0.00 (1.00) | 0.01 (-0.04, 0.07) |
| Southern oscillation index | 2; 4 | **0.11 (0.05)** | -0.04 (-0.25, 0.16) | 0.00 (1.00) | 0.01 (-0.01, 0.03) |

one (in the years 2010, 2011 and 2013) and a maximum of 14 admissions in the year 2017. The largest numbers of admissions were generally in October (range = 4–21 per month), while February–July showed minimal admissions (range = 0–5 per month; S1 Fig). The largest number of admissions during a single fortnight was 11, recorded in October 2009 and in August 2017. There was a decreasing trend through the most recent La Niña period (2010–2012), followed by an increasing trend from 2014 (Fig 2).

## Predictor model performances and estimated covariate associations

The best-fitting ARIMA$_{seasadj}$ model was an ARIMA(0,1,2) that included two predictor variables (SOI and moist vegetation$_{Q\text{-anomaly}}$; Table 2). In terms of prediction interval accuracy across the 2016 horizon, the GARCH$_{seasadj}$ was the best-performing of the four ensemble members. Optimized ensemble weights were 0.36 for the GARCH$_{seasadj}$, ARIMA, 0.33 for the GAM$_{raw}$, 0.31 for the ARIMA$_{seasadj}$ and 0 for the Prophet model. Coefficient estimates revealed several important environmental associations. Three models (GARCH$_{seasadj}$, GAM$_{raw}$ and Prophet) estimated a significant decrease in admissions with above average maximum temperatures during the previous month (lag 2 fortnights (4 weeks); Table 2). Note that GAM$_{raw}$ effects are presented as F statistics that do not reveal the directionality of the effect, only how 'nonlinear' it is. Directionality can be interpreted by visualising partial effects plots in S2 Fig. The ARIMA$_{seasadj}$ estimated that admissions significantly increased with increasing SOI (2-fortnight (4-week) lag) while the Prophet model estimated a significant increase in admissions with increasing values of the moist vegetation index (6-fortnight (12-week) lag; Table 2).

## Performances of ensemble forecasts

The ensemble outperformed the ETS benchmark when forecasting the 2017 tick season (Fig 4). Point errors for the ensemble had a median of 1.16 [Interquartile range (IQR): 0.48–2.08], lower at nearly all 2017 timepoints than the benchmark (median: 1.19; IQR: 0.54–2.64; Fig 5). Rolling MSIS interval errors were lower for the ensemble than the benchmark leading into the 2017 tick season across both horizons, though the benchmark performed equally well in the near-term when trained on all data up to the start of the season (September 2017; Fig 6). In the near-term, both models performed better when predicting the non-tick season (trained to April–May 2017), followed by increases in error as horizons began to cover the upcoming tick season (trained to June–July; Fig 6). As models gained access to observations from the tick season (trained to August–September), near-term errors decreased. In contrast, medium-term errors remained steady for both models through the year, with the predictor ensemble consistently outperforming the benchmark (Fig 6). When using particle filtering to assimilate observations and

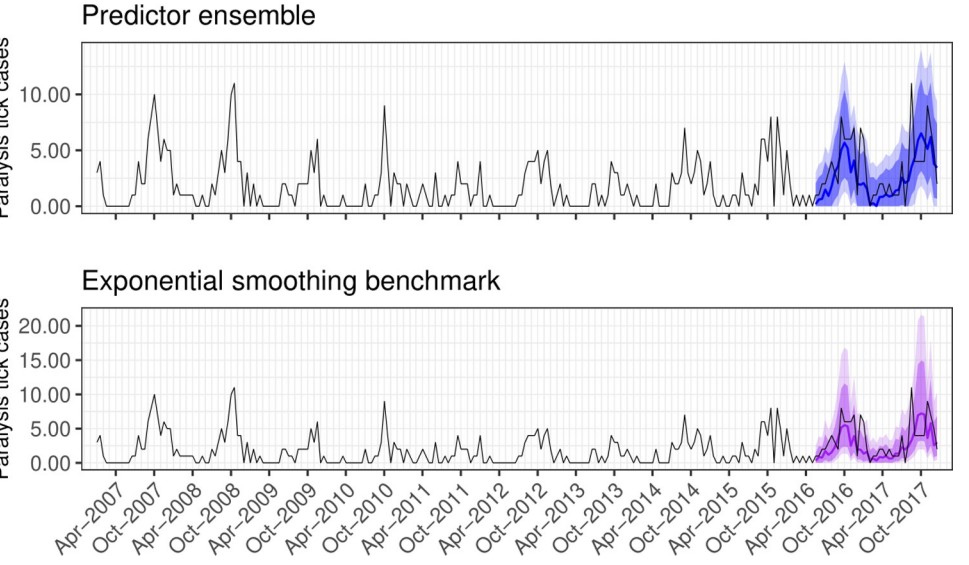

**Fig 4. Forecasts from competing models.** Forecasts of paralysis tick admissions (observed counts shown as the black line) generated by the predictor ensemble (blue shading) and the ETS benchmark model (purple shading). For both forecasts, dark shading shows 80% and light shading shows 95% prediction intervals. Note the difference in scales of the y-axes.

update the ARIMA$_{seasadj}$ and GARCH$_{seasadj}$ models (orange shading in Fig 6), rather than re-training all models on the growing dataset (blue shading in Fig 6), prediction intervals were further reduced across both horizons. See S3 and S4 Figs in Supporting Information for visualisations of how the ARIMA$_{seasadj}$ and GARCH$_{seasadj}$ models adapted to incoming observations. When adapting our model to clinic-level admissions, the particle filter again outperformed the ETS benchmark for all three clinic-level datasets (S5–S8 Figs). This difference in performance was most apparent for the northern and mid-latitude clinic series, while the southern clinic series was similarly predicted by the benchmark up to the beginning of the tick season in August–September, after which the particle filter became superior (S8 Fig).

## Discussion

Tick paralysis remains one of Australia's most important domestic animal health conditions [3,4,7]. Forecasts of tick paralysis cases that can guide public health awareness and clinical

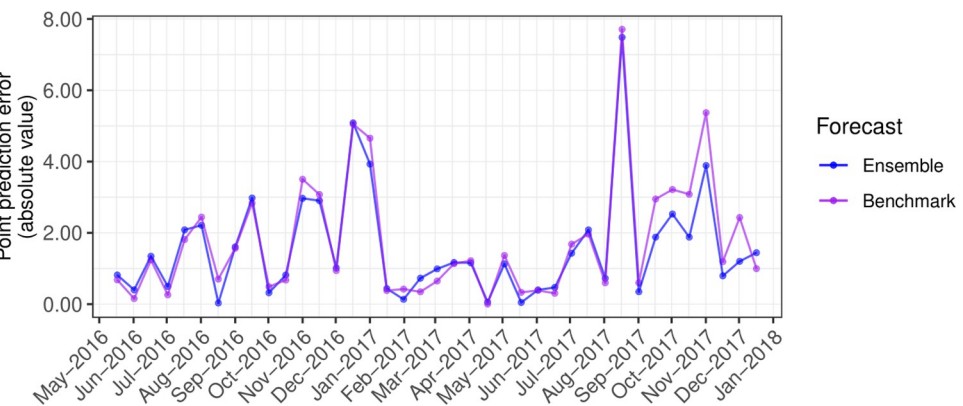

**Fig 5. Out-of-sample point prediction errors for the ensemble and ETS benchmark forecasts.**

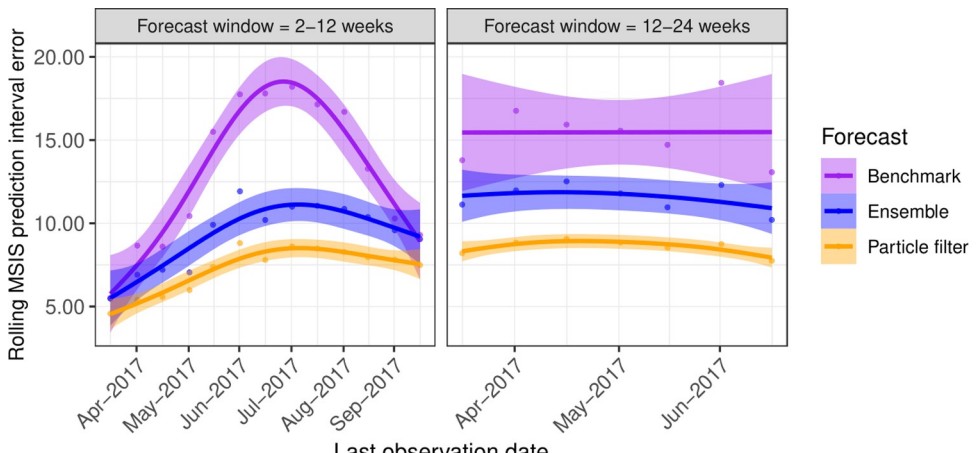

**Fig 6. Rolling prediction interval errors for the ETS benchmark, re-trained ensemble and particle filtered ensemble.** The benchmark (purple shading) and re-trained ensemble models (blue shading) were iteratively trained on the full dataset as observations became available to simulate a scenario in which models are continually re-calibrated to incoming data. The particle filter (orange shading) involved no retraining for the seasonally adjusted models (ARIMA$_{seasadj}$ and GARCH$_{seasadj}$), but instead used iterative assimilation of incoming observations via Sequential Monte Carlo. Lines and shaded areas show trends and 99% confidence intervals estimated using cubic regression splines.

management strategies could be crucial for reducing burdens of this often-fatal condition. Using a combination of statistical optimization and data assimilation methods, our ensemble framework generates reliable forecasts of tick paralysis admissions at horizons that would be appropriate for delivering early warnings to clinicians and pet owners in the Sunshine Coast. We show that our model consistently outperforms a field-leading benchmark by reducing both point and prediction interval errors, particularly when using particle filtering to automatically adapt forecasts over time. Crucially, our ensemble combines attractive properties of different time series and generative algorithms while uncovering important inferences about environmental drivers of paralysis tick exposure. We show that variables related to temperature anomalies, levels of moist vegetation and the Southern Oscillation Index show statistical associations with tick paralysis admissions in the Sunshine Coast, a finding that will have important implications for designing improved early warning systems.

The stable seasonality in tick paralysis admissions on the Sunshine Coast revealed by our study, with peak admissions between September and December, is supported by previous Australian research that recorded peak abundance of questing *Ixodes* ticks during spring and early summer [12,29]. The seasonality in domestic animal tick paralysis incidence in the East coast of Australia has been documented in earlier studies, with estimates that up to 65.5% of cases are reported from September to November [14,54]. The presence of stable seasonality in admissions is an important property that lends itself well to forecasting approaches that decompose the incidence time-series into seasonally adjusted components to identify environmental predictors of exposure. Our results indicate a possible negative effect of higher maximum temperature during the previous month on tick paralysis admissions. This finding is biologically plausible and in line with evidence that milder temperatures may be more conducive to increasing paralysis tick abundances and / or domestic dog exposures to ticks [11,27,55]. Our study also extends knowledge of tick paralysis dynamics in the study region by uncovering a non-linear trend in admissions that reached a trough following major rain and flooding events during the last La Niña phase (2010–2012) and began steadily rising afterward. We postulate that while moist vegetation is important for the survival and proliferation of

paralysis ticks, extreme rain and floods can impact tick abundances by altering habitats and reducing availabilities of vertebrate hosts. Indeed, previous research along the Danube River in Austria found that tick abundances were massively reduced following flood events, a pattern that the authors speculated was due to sediment influxes that indirectly influenced intermediate host availability [56]. It is also likely that during heavy rain and flooding events pet owners are less likely to allow their pets outside the household, reducing dog exposures to tick-prone environments, though this hypothesis requires testing.

There is a global need to develop decision-support tools that contribute to reduce community exposure and health impacts of vector-borne diseases, including tick paralysis and several tick-borne pathogens of public health significance [57,58]. Our ensemble model framework used advances in statistical forecasting and dynamic systems modelling to generate reliable forecasts of tick paralysis admissions at horizons that would be appropriate for delivering early warnings to clinicians and pet owners in the Sunshine Coast. This finding supports previous evidence that information regarding environmental covariates is useful for improving prediction of tick paralysis cases [55,59]. However, forecasts of tick paralysis admissions are likely to remain imperfect since clinical admissions due to tick paralysis are the result of many processes for which data are difficult to obtain, including the relative abundance and activities of dogs in tick-prone areas, microclimate and vegetation characteristics, wildlife distributions and social determinants such as veterinary health care seeking behavior of owners [4,10,13]. Nevertheless, the utility of our ensemble model was further amplified when using particle filtering, as opposed to model re-training, to adapt the forecast of tick paralysis incidence. Despite the wide recognition that filtering systems are paramount for tackling advanced multidimensional forecast problems such as weather prediction or moving object tracking [49,51,60], they are rarely employed for studying ecological time series [50,61]. Our results demonstrate that the particle filter produced by far the most accurate forecasts of the 2017 validation data across both horizons for the full dataset and for de-aggregated clinic-level datasets. We postulate that this improvement resulted from the particle filter's ability to break free from model constraints such as Gaussian errors and linear trends. Understanding these improvements was beyond our scope, however it is important to note that such investigation can be carried out in a hypothesis-driven framework using forecast uncertainty analyses to improve our understanding of the processes that drive ecological phenomena [50,62].

Several automated ecological forecasting frameworks exist, and we believe that our results show that tick paralysis admissions are worthy of such a system. This is particularly important given the lack of a monitoring system for Australian domestic animal health conditions. Australia's primary surveillance system for monitoring trends in companion animal tick paralysis over the past decade has been the Virbac Disease WatchDog (http://www.diseasewatchdog. org/). This passive system relied on clinicians around Australia to collate monthly cases and input data into the database. While this platform was commendable and helped identify tick paralysis risk factors [27], some drawbacks limited its utility as a warning system. Voluntary reporting was poor and reduced substantially over time [54]. In fact, reporting rates were so low that maintainers of the database concluded its records were likely not reflective of vector-borne disease outbreaks. The system was closed in 2017.

Welch et al [63] outline four primary stages necessary to implement a dynamic management tool: Acquisition, Prediction, Dissemination and Automation. While we focus primarily on prediction, the data and models we use would facilitate an automated forecast and analysis framework. For example, weather observations provided by LongPaddock (https://www. longpaddock.qld.gov.au/silo/ [31,32]) are updated nightly (one day latency) with new data from Australia's Bureau of Meteorology, while the rapidly growing VetCompass Australia database [64] collates clinical data from >300 veterinary clinics around Australia to facilitate

near real-time updating of domestic animal health analyses / forecasts. Ready access to data products, a version-controlled codebase with proper error logging, a reliance on particle filters to assimilate new observations and access to high performance compute clusters to log modelling / analysis jobs could form a streamlined workflow that self-initiates at regular frequencies to provide near-term updates on the progressing tick season. The design of a dynamic tick paralysis forecast tool would require inputs from stakeholders on the dissemination aspect, particularly to identify useful ways to visualize forecasts that overcome the common misperceptions about probabilities and uncertainty [65].

The results of our study are based on a modelling framework which could be improved in several ways. First, while we identified a single weighting scheme to reduce prediction interval errors across the 2016 validation period, the weight calculation step could be turned into a time-variant optimization problem if enough data is available to assess how well each ensemble member performs in certain forecast horizons or under certain conditions. In this way, the ensemble could be re-weighted over time to capture the inherent strengths of each ensemble member during different periods of volatility or growth. Similar approaches have been used successfully to forecast battery capacities and $CO_2$ emissions [66,67]. Second, we chose our predictor set as these variables were interpretable, they covered the entire temporal period of our tick paralysis data and, crucially, they are all continuously updated and stored in secure databases. Future iterations of our model could explore additional environmental and demographic predictors and could use time series feature engineering methods to uncover ways to improve our model's predictive capacity (i.e. by testing whether moving averages of predictors could improve forecasts, for example). We also did not use any post-estimation correction of estimated effect sizes as we were mainly interested in forecastability and less so on whether effects were 'significant', but studies designed for inference that wish to focus more on interpretation of estimated effects would need to consider post-estimation adjustments to account for our heuristic model selection strategy. Finally, we were only able to test our model's performance using a single year of holdout data (2017). Future studies could continuously update our model and compare it to competing models over time to inspect model failures and make incremental improvements. This may be especially necessary when modelling tick paralysis or tick densities in general, as it is not uncommon for tick-borne disease cases to demonstrate 2–3 years cycles driven by the population dynamics of ticks [68]. There are a range of models that can incorporate multiple seasonalities, including the GAM and Prophet models we used here, and future work should investigate whether inclusion of these cycles improves near-term forecasts.

Despite these limitations, our results demonstrate the utility and the potential for our near-term forecasting model to inform the public health management of tick paralysis in a high-risk region. Our study sits within the growing field of near-term ecological forecasting [20,21,50] by providing an example for how forecast systems can be developed to study not only companion animal tick paralysis but also with wider applicability to manage veterinary vector-borne diseases more broadly.

## Supporting information

**S1 Fig. Boxplots of monthly tick paralysis admission counts to Sunshine Coast GreenCross veterinary clinics across the study period (2007–2017).**
(TIFF)

**S2 Fig. GAM_{raw} partial effect smooth function plots.**
(TIFF)

**S3 Fig. ARIMA particle filter visualization.** Forecasts of seasonally adjusted paralysis tick admissions (truth shown as the black line) generated by the original ARIMA$_{seasadj}$ (blue shading) and the ARIMA$_{seasadj}$ following particle filtering assimilation of the first six months of observations in the out-of-sample validation set (orange shading). For both forecasts, dark coloured shading shows 80% and light shading shows 95% prediction intervals.
(TIFF)

**S4 Fig. GARCH Particle filter visualization.** Forecasts of seasonally adjusted paralysis tick admissions (the true observations are shown as the black line) generated by the original GARCH$_{seasadj}$ (blue shading) and the GARCH$_{seasadj}$ following particle filtering assimilation of the first six months of observations in the out-of-sample validation set (orange shading). For both forecasts, dark coloured shading shows 80% and light shading shows 95% prediction intervals.
(TIFF)

**S5 Fig. STL decompositions for each of the three clinic-level paralysis tick admissions datasets.** The trend components are centred ($x_{centre} = x - mean(x)$) to facilitate simpler comparisons of their temporal dynamics.
(TIFF)

**S6 Fig. Northern clinic rolling interval errors.** Rolling prediction interval errors for the ETS benchmark (purple shading) and particle filtered ensemble (orange shading) applied to paralysis tick admissions for GreenCross clinics in the northern region. The benchmark model was iteratively retrained on the full dataset as observations became available to simulate a scenario in which models are continually re-calibrated to incoming data. The particle filter involved no retraining for the seasonally adjusted models (ARIMA$_{seasadj}$ and GARCH$_{seasadj}$), but instead used iterative assimilation of incoming observations via Sequential Monte Carlo. Lines and shaded areas show trends and 99% confidence intervals estimated using cubic regression splines.
(TIFF)

**S7 Fig. Mid-latitude clinic rolling interval errors.** Rolling prediction interval errors for the ETS benchmark (purple shading) and particle filtered ensemble (orange shading) applied to paralysis tick admissions for the mid-latitude GreenCross clinics. The benchmark model was iteratively retrained on the full dataset as observations became available to simulate a scenario in which models are continually re-calibrated to incoming data. The particle filter involved no retraining for the seasonally adjusted (ARIMA$_{seasadj}$ and GARCH$_{seasadj}$), but instead used iterative assimilation of incoming observations via Sequential Monte Carlo. Lines and shaded areas show trends and 99% confidence intervals estimated using cubic regression splines.
(TIFF)

**S8 Fig. Southern clinic rolling interval errors.** Rolling prediction interval errors for the ETS benchmark (purple shading) and particle filtered ensemble (orange shading) applied to paralysis tick admissions for the GreenCross clinic in the southern region. The benchmark model was iteratively retrained on the full dataset as observations became available to simulate a scenario in which models are continually re-calibrated to incoming data. The particle filter involved no retraining for the seasonally adjusted (ARIMA$_{seasadj}$ and GARCH$_{seasadj}$), but instead used iterative assimilation of incoming observations via Sequential Monte Carlo. Lines and shaded areas show trends and 99% confidence intervals estimated using cubic regression splines.
(TIFF)

**S1 Table. Assumptions of predictor models used to forecast paralysis tick admission cases to Sunshine Coast veterinary clinics and build the ensemble forecast.** ARIMA, Autoregressive Integrated Moving Average; GARCH; Generalized Autoregressive Conditional Heteroskedasticity; GAM, Generalized Additive Model; ETS, exponential smoothing.
(DOCX)

**S1 Data. Data and R code to replicate ensemble modelling and particle filtering and associated figures. Model results from this study are included in the *Intermediate_outputs* folder.**
(RAR)

## Acknowledgments

We thank participating Greencross Veterinary clinics on the Sunshine Coast for their generous provision of access to primary clinical data.

## Author Contributions

**Conceptualization:** Nicholas J. Clark.

**Data curation:** Nicholas J. Clark, Guyan Weerasinghe, Ricardo J. Soares Magalhães.

**Formal analysis:** Nicholas J. Clark, Tatiana Proboste.

**Investigation:** Nicholas J. Clark.

**Project administration:** Guyan Weerasinghe, Ricardo J. Soares Magalhães.

**Validation:** Nicholas J. Clark.

**Visualization:** Nicholas J. Clark, Tatiana Proboste, Ricardo J. Soares Magalhães.

**Writing – original draft:** Nicholas J. Clark.

**Writing – review & editing:** Nicholas J. Clark, Tatiana Proboste, Guyan Weerasinghe, Ricardo J. Soares Magalhães.

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
