## [Decision Letter · Decision Letter 0]

13 Oct 2021

Dear Dr Clark,

Thank you very much for submitting your manuscript "Near-term forecasting of companion animal tick paralysis incidence: an iterative ensemble model" for consideration at PLOS Computational Biology.

As with all papers reviewed by the journal, your manuscript was reviewed by members of the editorial board and by several independent reviewers. In light of the reviews (below this email), we would like to invite the resubmission of a significantly-revised version that takes into account the reviewers' comments.

We cannot make any decision about publication until we have seen the revised manuscript and your response to the reviewers' comments. Your revised manuscript is also likely to be sent to reviewers for further evaluation.

Sincerely,

Benjamin Muir Althouse

Associate Editor

PLOS Computational Biology

Virginia Pitzer

Deputy Editor-in-Chief

PLOS Computational Biology

Reviewer's Responses to Questions

**Comments to the Authors:**

Reviewer #1: Here the authors have built a time series forecasting model using incidence counts of tick paralysis cases. The model is a weighted ensemble of several well known time series models which outperformed a baseline model, providing a proof of concept for a potential forecast for tick awareness to promote public health.

This paper is excellent. It is clear, well written, and incorporates all best practices of near-term time series forecasting. Namely that the model uses an ensemble where individual models are weighted by their relative performance, it incorporates new observations via assimilation, compares against a relatively simple base model, and evaluates both point prediction and confidence interval performance. In its current form this manuscript is certainly publishable, and is novel for bringing together the above described aspects of time series forecast and presents everything in supplemental R code.

But I feel it is lacking in that, similar to many forecasting studies, it discusses only the modelling component and not the needed logistics for actually implementing it and making tick forecasts a usable resource. Welch et al 2019 points out the four primary parts of this process, and the current manuscript only addresses one of those. As researchers read this manuscript for the modelling aspect, it would benefit them to be exposed to the bigger picture and how those models fit into and affect a larger system. A “next steps” in the discussion would suffice for this, and would help push the field of ecological forecasting forward beyond just building and evaluating the core models. Some potential points would be:

- How feasible is it to get real-time tick counts from clinics to take advantage of the data assimilation? The same goes for the various weather drivers which are part of the model.

- Is presenting predicted tick counts a useful way to disseminate this information? Would the public actually change their actions based on that information? This ties directly to the interpretability discussed throughout the manuscript. For example a potentially better way to present the forecasts could be color coded symbols representing discrete bins of current tick activity, similar to color coded bush fire dangers. See the following papers on forecast presentation for more discussions on this, which are largely from the weather community as it’s an understudied aspect in ecology: Morss et al 2008, Gerst et al 2020, Hartmann et al 2002, Raftery 2016

- If this was implemented and was successful in reducing tick bites, the model would then become ineffective since actual bites, and not risk of tick bites, is the predicted response variable. What could be done then to keep it operational?

Some other minor points:

L104: I would not consider three studies a plethora. But you can add Dormann et al 2018, Tebaldi 2007, and Thomas et al 2020 to make it close.

L158: fitting the ARIMA and GARCH models to the trend + residuals is essentially the same as fitting to the anomaly of the time series, which is appropriate. I would not consider this approach a “state space” model though, since a hidden state is not being estimated at this step (though it is applicable to the data assimilation step). Thus please remove, or better clarify, the state space references (eg. L 366, 388, 470, 739, 757)

L771: please also deposit the github repository on either zenodo or figshare since github is not a permanent archive.

Dormann, Carsten F., et al. 2018. “Model Averaging in Ecology: A Review of Bayesian, Information-Theoretic, and Tactical Approaches for Predictive Inference.” Ecological Monographs 0 (0): 1–20. https://doi.org/10.1002/ecm.1309.

Gerst, Michael D., et al. 2020. “Using Visualization Science to Improve Expert and Public Understanding of Probabilistic Temperature and Precipitation Outlooks.” Weather, Climate, and Society 12 (1): 117–33. https://doi.org/10.1175/WCAS-D-18-0094.1.

Morss, Rebecca E., Julie L. Demuth, and Jeffrey K. Lazo. 2008. “Communicating Uncertainty in Weather Forecasts: A Survey of the U.S. Public.” Weather and Forecasting 23 (5): 974–91. https://doi.org/10.1175/2008WAF2007088.1.

Hartmann, Holly C., Thomas C. Pagano, S. Sorooshian, and R. Bales. 2002. “Confidence Builders: Evaluating Seasonal Climate Forecasts from User Perspectives.” Bulletin of the American Meteorological Society 83 (5): 683–98. https://doi.org/10.1175/1520-0477(2002)083<0683:CBESCF>2.3.CO;2.

Raftery, Adrian E. 2016. “Use and Communication of Probabilistic Forecasts.” Statistical Analysis and Data Mining: The ASA Data Science Journal 9 (6): 397–410. https://doi.org/10.1002/sam.11302.

Tebaldi, C. and Knutti, R., 2007. The use of the multi-model ensemble in probabilistic climate projections. Philosophical transactions of the royal society A: mathematical, physical and engineering sciences, https://doi.org/10.1098/rsta.2007.2076

Thomas, R Quinn, Renato J Figueiredo, Vahid Daneshmand, Bethany J Bookout, Laura K Puckett, and Cayelan C Carey. 2020. “A Near‐Term Iterative Forecasting System Successfully Predicts Reservoir Hydrodynamics and Partitions Uncertainty in Real Time.” Water Resources Research 56 (11). https://doi.org/10.1029/2019WR026138.

Welch, Heather, Elliott L. Hazen, Steven J. Bograd, Michael G. Jacox, Stephanie Brodie, Dale Robinson, Kylie L. Scales, Lynn Dewitt, and Rebecca Lewison. 2019. “Practical Considerations for Operationalizing Dynamic Management Tools.” Edited by Annabel Smith. Journal of Applied Ecology 56 (2): 459–69. https://doi.org/10.1111/1365-2664.13281.

Reviewer #2: The manuscript presents an interesting methodology for forecasting cases of tick paralysis that is very relevant to the field of vector-borne disease research with useful applications to other vector-borne pathogens. The paper is well written and explains the methodology in detail and is easy to follow. I have some questions that might help clarify some points.

1. While I appreciate the description of the models in detail, I would like to know the rationale for selecting some of them and not others… for example: why not a random forest model?

2. Why were the temporal lags expressed in weeks when cases were aggregated by fortnights? Shouldn’t they be on the same scale?

3. Apart from the overall trend, it’s not uncommon to have 2-3 years cycles in tick-borne disease cases driven by the population dynamics of ticks. I guess in a way you can visualize it in the trend, but that also includes the long-term trend… based on the seasonal-trend decomposition based on Loess smoothing could you capture those cycles and separate it from the long term trend (say, driven by ISO)? Could you maybe observe that by analyzing the temporal autocorrelation of the “reminder”? I guess that is what you where adjusting for in the GARCH model?

4. What was the spatial resolution of the predictors? I understand the resolution in the original raster files but how were they introduced in the models? Did you use buffers around the clinics? Did you average across the whole study area? Same with the temporal data… did you use the fortnight or the actual date of diagnosis for each case? The only one that is obvious to me is the landcover which was obtained yearly for the whole study area as indicated in the text.

5. When selecting the best model in the ARIMA model, did you have competing models? Did you account in any way for model selection uncertainty?

6. Line 265-266: “This model [GARCH model] was an appropriate alternative to the ARIMA as variations in the reminder series showed alternative periods of volatility.” Did you try an autocorrelation function for the reminder? See point 3.

7. Line 269-271: Could you explain why regularized double exponential priors were used?

8. For all models (except for the ARIMA model that is explained): how where the lags selected?

9. Lines 314-315 (Prophet model): The three predictors were selected by testing in each on its own, why not used the variables for which you have some evidenced based on the previous models? Just again thinking about model selection uncertainty.

10. I don’t think I completely follow the Prophet model, how does it assimilate new data?

11. Table 2 has a few problems: check the signs of the coefficients (for example, in the GARCH model the point estimate is not included in the CI for Maximum temp).

12. Line 423: “Three models indicated a significant decrease in admissions with above average maximum temperatures…” But for the GAM model we observe that the coefficient is in the opposite direction compared with the two other mentioned models.

13. Line 507-509: “This is supported by the Prophet model’s estimate of increased admissions with increasing SOI, etc.” How is the non-linear trend of this model accounting for a effects of extreme rainfall? I would be very careful about interpreting the model results in general… you are not looking at inference but forecasting…

14. Line 523: I do agree that exposure data based on demographic and behavioral data are hard to obtain, but I wouldn’t necessarily call it an stochastic process. It would be interesting to control for admissions in the clinics overall (maybe considering another disease that has no seasonality and it’s not related to tick paralysis).

Minor revisions:

Line 267: ARIMA(0,2) instead of ARMA?

Line 412-413:When it says October 2008, shouldn’t it be 2009? The peak in Fig 2 seems higher in 2009

Lines 414-415: Shouldn’t it say 2014 instead of 2014? That would match the Fig 2 trend.

**Have the authors made all data and (if applicable) computational code underlying the findings in their manuscript fully available?**

Reviewer #1: **No: **The primary modelling code was made available and the authors state it and the raw data (minus the tick incidence counts due to privacy reasons) will be available on publication.

Reviewer #2: Yes

PLOS authors have the option to publish the peer review history of their article (what does this mean?). If published, this will include your full peer review and any attached files.

Reviewer #1: No

Reviewer #2: No
---

## [Decision Letter · Decision Letter 1]

27 Jan 2022

Dear Dr Clark,

We are pleased to inform you that your manuscript 'Near-term forecasting of companion animal tick paralysis incidence: an iterative ensemble model' has been provisionally accepted for publication in PLOS Computational Biology.

Best regards,

Benjamin Althouse

Associate Editor

PLOS Computational Biology

Virginia Pitzer

Deputy Editor-in-Chief

PLOS Computational Biology

Reviewer's Responses to Questions

**Comments to the Authors:**

Reviewer #1: The authors have addressed all my concerns and I can recommend this paper for publication. This will be an excellent addition to the ecological forecasting literature.

Reviewer #2: The authors have successfully addressed all my comments and questions. The reviewed manuscript is more clear on the methods, some of the modeling decision and their rationale, and implications of their results. I don't have any more comments.

**Have the authors made all data and (if applicable) computational code underlying the findings in their manuscript fully available?**

Reviewer #1: **No: **Figshare link to be available upon acceptance

Reviewer #2: Yes

PLOS authors have the option to publish the peer review history of their article (what does this mean?). If published, this will include your full peer review and any attached files.

Reviewer #1: **Yes: **Shawn D Taylor

Reviewer #2: No

---

## [Editor Report · Acceptance letter]

11 Feb 2022

PCOMPBIOL-D-21-00714R1 

Near-term forecasting of companion animal tick paralysis incidence: an iterative ensemble model

Dear Dr Clark,

I am pleased to inform you that your manuscript has been formally accepted for publication in PLOS Computational Biology. Your manuscript is now with our production department and you will be notified of the publication date in due course.

With kind regards,

Orsolya Voros
